# STRATEGIES AND IMPACT OF LEARNING CURVE ESTIMATION FOR CNN-BASED IMAGE CLASSIFICATION

## ABSTRACT

Learning curves are a measure for how the performance of machine learning models improves given a certain volume of training data. Over a wide variety of applications and models it was observed that learning curves follow – to a large extent – a power law behavior. This makes the performance of different models for a given task somewhat predictable and opens the opportunity to reduce the training time for practitioners, who are exploring the space of possible models and hyperparameters for the problem at hand. By estimating the learning curve of a model from training on small subsets of data only the best models need to be considered for training on the full dataset. How to choose subset sizes and how often to sample models on these to obtain estimates is however not researched. Given that the goal is to reduce overall training time strategies are needed that sample the performance in a time-efficient way and yet leads to accurate learning curve estimates. In this paper we formulate the framework for these strategies and propose several strategies. Further we evaluate the strategies for simulated learning curves and in experiments with popular datasets and models for image classification tasks.

## 1 INTRODUCTION

In recent years deep learning (DL) models have led to impressive advancements in a wide variety of fields, such as self-driving cars, medicine, and digital agriculture, to just name a few. These models are fundamentally data-driven, where the performance of a trained model correlates with the quality, but (seemingly) mostly with the quantity of data. At the same time the training time (and the costs of training) scales with the quantity of data. Besides handling these large datasets, practitioners usually have a wide choice of models at their disposal, each of which can be further tuned by adjusting its hyperparameters. Thus, to solve a specific problem with a given dataset many models must be trained and tested until one is found that performs to our expectations. To shorten this time-intensive process one solution is to train models on a small training set first, assuming that models that outperform others will continue to do so when trained on the full training set. This, however, is generally not true, as can be seen, for example, in Table 1.

Learning curves, also known as neural scaling laws (e.g., Hestness et al. (2017), are the description of how a model's performance increases when it is trained on more training data. A situation in which model $M_1$ outperforms another model $M_2$ when trained on a small training set but is being outperformed when trained on a larger training set can thus be identified by the respective learning curves crossing each other. Learning curves give us more accurate information in which model to invest our time when it comes to training on the full dataset.

Unfortunately, due to the complexity of DL models, an exact mathematical formulation of learning curves is not known and might be out of reach for all but the simplest models. However, through empirical observations it was observed that learning curves can be described using fairly simple functions (Rosenfeld et al. (2019)). Amongst these a power law relationship between loss and training volume of the form $\eta(x) = x^{\theta_1}\theta_2$ is a popular choice, where $x$ represents the amount of training data used. To answer which model will perform best the next natural step must then be to estimate the parameters $\theta_1$ and $\theta_2$ for each model class. This in turn requires training on at least some data volumes and the question becomes which volumes to train on and how often. Here, we trade off

Table 1: Performance estimates based on single training volume

| Models | Acc. 1.8K | Acc. 90K | Rank 90K |
|---|---|---|---|
| DenseNet169 | 72.9% | 85.7% | 5 |
| DenseNet201 | 72.0% | 85.3% | 6 |
| ResNet101 | 71.2% | 87.1% | 1 |
| ResNet50 | 69.1% | 86.0% | 4 |
| MobileNetV3Large | 69.0% | 86.1% | 3 |
| ResNet152 | 68.3% | 86.9% | 2 |
| DenseNet121 | 68.2% | 81.3% | 7 |
| VGG16 | 64.2% | 76.1% | 10 |
| VGG19 | 64.1% | 76.8% | 9 |
| MobileNetV3Small | 63.5% | 79.1% | 8 |
| Xception | 61.5% | 68.8% | 13 |
| NASNetMobile | 60.0% | 75.4% | 11 |
| InceptionV3 | 59.8% | 67.7% | 14 |
| NASNetLarge | 56.3% | 69.4% | 12 |

Table 1: Values of the OOD accuracy on a validation test-set for different models trained on a small training subset of 1,800 images and for training the models on the full training set of 90,000 images of the Plant Dataset (see Section 5. The best three performing models, when trained on the full training set, and the two best performing models, when trained on the subset only, are highlighted in blue and red, respectively.

accuracy of the learning curve fit with the costs of estimating the learning curves in the first place (an effort we could have spent on just train different models on the full training set instead). In this paper we discuss several sampling strategies and evaluate them with respect to training time saved. The performance of a sampling strategy is in this case the difference in loss when using the model that was *predicted* to perform best versus the model that actually performs best. Our evaluations include common convolutional neural network architectures for image classification on three different datasets. In addition to that, we propose a model for learning curves to simulate learning outcomes, which allows us to evaluate the sampling strategies on an even wider scope.

Overall, the contributions of this paper are:

- We introduce the concept of fitting learning curves from samples on small training volumes and the accompanying mathematical notation.
- We propose a model for learning curves that captures its three main regions: First, when training volumes are too small and the model fails to learn; second, the region in which the learning curve descents along a power law; and third, when the learning curve approaches an irreducible error. This allows us to simulate training outcomes on different training volumes in a fast way.
- We describe and investigate several sampling strategies for the fitting of learning curves from training on small volumes. These strategies are evaluated on the simulated learning curves from our model, as well as in three different application scenarios of common convolutional neural network architectures applied to real world data.

RELATED WORK

Our work builds upon the insights on neural scaling laws – or **learning curves** – which have been gathered in the last years with respect to deep learning models. Early application of learning curves in machine learning can be found for example in Mukherjee et al. (2003) and Figueroa et al. (2012). Both works tackle the goal of how to estimate the amount of training data that is needed to train a model to a performance target. With the advent of deep learning models also a description of their learning curves was investigated. The authors of Hestness et al. (2017) laid important groundwork by empirically measuring and fitting learning curves over different machine learning domains. A

deeper investigation into the parametrization of learning curves was performed in Rosenfeld et al. (2019). A comprehensive review of learning curves, including empirical and theoretical arguments for power-law shapes, as well as ill-behaved learning curves, is given in Viering & Loog (2023). On the side of utilizing learning curves for data collection we want to mention Mahmood et al. (2022a), Mahmood et al. (2022b), which are the closely related to our results. They investigate how learning curve fits can answer how much more data is needed to reach a goal performance. The difference to our work is in this case, that we assume that a dataset is already collected, and we rather want to find the best performing model in a quick manner. In this sense our work is complementary to Mahmood et al. (2022a) and Mahmood et al. (2022b). In Hoiem et al. (2021) important aspects of training models, such as pretraining, choice of architecture, and data augmentation, are investigated with the help of learning curves. Our work differs from Hoiem et al. (2021) by considering sampling strategies for the learning curve estimation, especially the costs of sampling (i.e., training on subsets) and the performance achieved, when choosing models accordingly.

The idea of progressive sampling connects our work with the areas of **active learning** (Cohn et al. (1996), Settles (2009)) and semi-supervised learning (Chapelle et al. (2006)), in which additional data is added (and labelled) into the training set iteratively as the model is training (e.g., Wang et al. (2017), Gal et al. (2017), Haut et al. (2018), Sener & Savarese (2018)). This is often performed with a given target volume of training data in mind. In our work we do reduce the amount of data used in model training, but we do not grow the training set by investigating which data points would be best to include. Indeed, all our smaller training sets are just a class-balanced random selection of the full training set. Again, we see our work complementary; indeed, we could follow the same strategies outlined in this paper but replace the random selection process by active learning.

Our work is part of **neural architecture search** (see Elsken et al. (2019)) and **performance prediction**. Determining power law behavior for learning curves reaches back much further than recent deep models. In Frey & Fisher (1999) and Gu et al. (2001) the authors evaluate a power law to be the best fit for learning curves of C4.5 decision trees and logistic discrimination models. The authors of Kolachina et al. (2012) determined the power law to be the best fit in their application scenario (statistical machine translation) as well. Another definition for learning curves in DL is the performance of the model as it progresses through epochs of training. Works under this definition of learning curves include Domhan et al. (2015), Klein et al. (2016), and Baker et al. (2017), which like our work have the goal of finding the best models or set of hyperparameters in shorter training time. While the aforementioned works use probabilistic models to extrapolate the learning curve, the work of Rawal & Miikkulainen (2018) uses a LSTM-network instead to predict a model's performance when its training has finished. Besides a different definition for learning curve our work also differs from these by exploring strategies on which data volumes to evaluate.

## 2 NOTATIONS

For the scope of this paper and unless noted otherwise, when we mention a model's *performance*, we mean the model's top-1 accuracy loss on a held out test set $\mathbb{T}$. We also call this the out of distribution (OOD) loss. Further, the full training set is often called the *target training set* and the number of samples in it the *target volume*. The task our machine learning models will learn is a mapping from a space of possible samples $\mathbb{A}$ to a set of labels $\mathbb{B}$. We denote the target training set by $\mathbb{S} \subset \mathbb{A}$ with $|\mathbb{S}| = x_N$ being the *target volume*. Let $\mathbb{S}_1 \subset \mathbb{S}_2 \subset \ldots \subset \mathbb{S}_n \subset \mathbb{S}$ be a sequence of increasing subsets of training samples and let $\boldsymbol{x} = (x_1, \ldots, x_n)$ be the respective training volumes, i.e., $x_i = |\mathbb{S}_i|$. We will use the terms training subset and training volumes interchangeably. We consider a family of models $\mathcal{M}$, where each $M \in \mathcal{M}$ is a function

$$M : \mathbb{A} \to \mathbb{B}.$$

The form of $M$ depends on many factors, such as model architecture, weight initialization, size and selection of samples and validation sets, and training procedure. Once model classes $\{M_1, \ldots, M_m\} \subset \mathcal{M}$ have been selected, we can train them on any training subsets $\mathbb{S}_i$ and measure their OOD performance. For brevity we call such a model $\mathbb{S}_i$-*trained* and denote it by $M_{i,j}$. Formally, we define the training function $\tau$ by

$$\tau : \{\mathbb{S}_1, \ldots, \mathbb{S}_n\} \times \{M_1, \ldots, M_m\} \to \epsilon(\mathbb{T})$$
$$(\mathbb{S}_i, M_j) \mapsto y_{i,j}$$

Since, there is usually randomness involved in the training process (e.g., the order in which the samples are being processed or the initialization of model weights), it is more useful to define the function $\tau$ as a random variable, such that the resulting $y_{i,j}$ is just one realisation of it. Thus, it is useful to sample $y_{i,j}$ more than once and extend the notation to $y_{i,j}^{(r)}$ denoting the $r$-th repetition of training the model. More repetitions give us a more accurate estimate for $y_{i,j}$ on one side, but also require more training time on the other side. Finding a good value for the number of repetitions for each training volume is thus one of the main challenges for estimating the performance of $M_{N,j}$.

The goal of training models on comparably small $\mathbb{S}_i$ is to estimate their performance on the full training set via the help of learning curves. We define a learning curve as a function $\eta$ that maps training volumes to OOD model performance[1]:

$$\eta : \mathbb{N} \to \mathbb{R}^+$$
$$x \mapsto x^{\theta_1} \cdot \theta_2$$

We note at this point that the true learning curve, given a non-trivial model and data distribution, is unknown. Indeed, even its power law parametrization as given here, is subject to research. Overall, for DL models we can consider the true learning curve to be unobtainable. Consequently, the goal is to estimate the learning curve parameters $\theta_1, \theta_2$ from training outcomes. If we fix the type of model to use by $M_j$ and train it over several training volumes $\mathbb{S}_i$ (once for each subset) we get a set of pairs $\{(x_i, y_{i,j})\}_i$ onto which we can fit a learning curve $\eta_j$ that describes the performance of model $M_j$ with respect to its training volume. In the remainder of this paper we denote this process as *sampling* model $j$ on volume $i$. Applying a non-linear least-squares fit to the pairs $\{(x_i, y_{i,j})\}_i$ results then in fitted parameters $\hat{\theta} = (\hat{\theta}_1, \hat{\theta}_2)$ and an estimated learning curve $\eta_j(\cdot, \hat{\theta})$ or in short $\hat{\eta}_j$. In case that individual subsets have been resampled, i.e., we have $(y_{i,j}^{(1)}, y_{i,j}^{(2)}, \ldots, y_{i,j}^{(l_{i,j})})$ the learning curve is not fit to the individual samples, but to their average instead; in short, we use

$$(x_i, \bar{y}_{i,j}) := \left( x_i, \frac{1}{l_{i,j}} \sum_{r=1}^{l_{i,j}} y_{i,j}^{(r)} \right). \tag{1}$$

We will use $\hat{\eta}_j(x_N)$ to estimate $M_{N,j}$. The goal is to know which $\mathbb{S}$-trained model will have the best OOD loss before performing the respective training.

The training of any model on any training subset requires computational effort. Depending on the application and environment these costs can come in different forms, for example, training time, required energy, or monetary costs. In the following formulation we just use the abstract term "costs" and assume a linear relationship between these costs and training volume[2]. We can now distinguish two different costs in training times. First, the costs of sampling models on $\mathbb{S}_1, \ldots, \mathbb{S}_n$ to obtain a learning curve fit; we denote these costs by $C_s$. Second, the costs of training $k$ models, selected according to $\hat{\eta}_j(x_N)$, on $\mathbb{S}$. These costs are denoted by $C_t$. We further introduce the costs of training each models $M_j$ on the target volume and denote it by $C_N$. The costs $C_N$ just represent the costs of applying a brute-force method to find $\min_j \{y_{N,j}\}$.

For simplicity of describing our results we further assume that the linear relationship between training volume and costs is the same amongst models. This assumption does generally not hold, the principles of our methodology remain however the same. Further, in our applications we have observed that this assumption is reasonably accurate. We thus can express the cost $C_s$ as

$$C_s = \gamma \sum_{j=1}^{m} \sum_{i=1}^{n} l_{i,j} x_i, \tag{2}$$

---

[1]This power law is a common parametrization for learning curves and is based on observations of learning curves over a wide variety of applications and models. For the scope of this paper, we adopt power law parametrization, but want to point out, that other parametrizations have been proposed, see for example Rosenfeld et al. (2019)

[2]In practice the costs are also a random variable and should be estimated as well. For the present discussion we however avoid a constant reminder that we should consider *expected* costs.

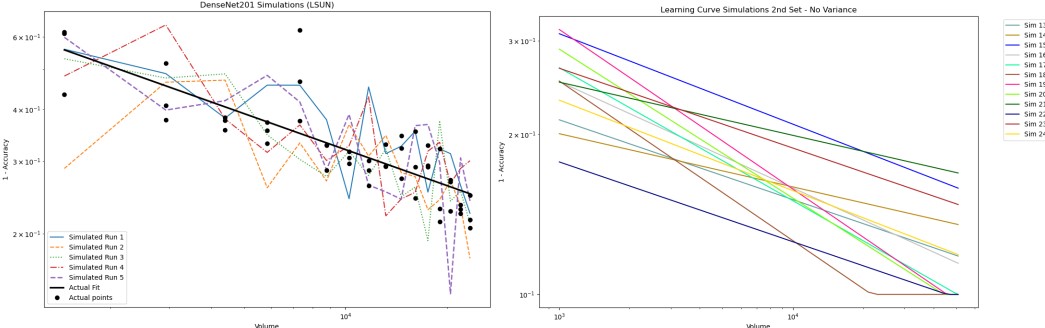

Figure 1: **Left:** Log-log plot of the learning curve of DenseNet201 that was trained on LSUN. The black dots each represent one training run of the model, there are three for each volume. Each line represents a simulated training run of on LSUN. **Right:** Log-log plot of the simulated learning curves without variance terms.

where $\gamma$ is some proportionality constant. Similarly, we have

$$C_t = \gamma \sum_{j \in j_1, \ldots, j_k} x_N = \gamma k x_N \tag{3}$$

$$C_N = \gamma \sum_{j=1}^{m} x_N = \gamma m x_N. \tag{4}$$

Eventually, the proportionality constant cancels out, since we report total costs as

$$C_s + C_t = x \cdot C_N \tag{5}$$

and $x$ only depends on the values of the $l_{i,j}$, $k$, and $x_1, \ldots, x_n, x_N$.

## 3 SAMPLING STRATEGIES

We now give an overview on several strategies that could be employed to select $k$ candidate models that will be trained on $\mathbb{S}$. The core problem is how to set the $l_{i,j}$ such that we get a good estimate on $y_{N,j}$, but also keep $C_s$ small.

One of the simplest strategies we can follow is to ensure that each combination of model class and volume is being trained equally often. This means we choose volumes $x_{i_1}, \ldots, x_{i_s}$, an integer $b$, and set

$$l_{i,j} = \begin{cases} b & \text{if } i \in \{i_1, \ldots, i_s\} \\ 0 & \text{else} \end{cases} \tag{6}$$

We can then further distinguish special cases where $\{i_1, \ldots, i_s\}$ represent only two volumes (i.e., $\{i_1, i_2\}$) or even a single volume (i.e., just $\{i_1\}$). In the former case we can compute the learning curve parameters of Equation 1 by solving a simple system of equations. In the later the learning curve equation is over-parameterized and we resort instead of using $y_{1,j}$ directly to determine the $k$ selected models.

## 4 SIMULATION OF LEARNING CURVES

To evaluate the above mentioned strategies, we apply them to experiments on three different datasets, as well as simulated results. For the latter, we create a simple model for learning curves that follows the form proposed by Hestness et al. (2017), therein three regions, the small data region, power-law region, and the irreducible error region, are identified. Accordingly, each classifier's performance over training volumes is split into three parts: First, training results are close to the classifier randomly guessing until a certain threshold $v_0$ of training samples is reached; second, when training on $v_0$ or more training samples the classifier's loss descends along a power-law learning curve until it

converges towards an irreducible loss at training volume size $v_\omega$; third, for training volumes $v_\omega$ and larger the classifier does not improve any further. To decrease the loss beyond this threshold would require a change of the model class (e.g., using a more complex model architecture).

To simulate accuracy loss we propose the following, where $v$ is the training volume[3]:

$$\eta(v; c, \delta, \theta_2, \theta_1, \sigma_M) = \begin{cases} 1 - c + \varepsilon_{v_0} & \text{for } 0 \leq v \leq v_0 \\ [\theta_2 \cdot v^{\theta_1} + \varepsilon_v + \varepsilon_M(v)]^+ & \text{for } v_0 < v < v_\omega \\ [\delta + \varepsilon_v + \varepsilon_M(v)]^+ & \text{for } v_\omega \leq v \end{cases} \tag{7}$$

The parameters are interpreted as follows: $c$ is the chance of guessing the correct class correctly, i.e., $c = (\text{number of classes})^{-1}$. The minimum loss the model can reach is given by $\delta$. The parameters $\theta_1$ and $\theta_2$ relate as before to the power law parametrization. The volumes $v_0$ and $v_\omega$ determine the change of regions from the small data region to the power-law region and to the irreducible error region, respectively. The terms $\varepsilon_v$ and $\varepsilon_M$ are variance terms defined as follows. The variance between separate runs of training a single model on a single volume is realized by $\varepsilon_v$. This variance results from the random chance by which the classifier can give the correct prediction, either by successfully extracting the relevant features from the sample or by random chance. We assume a normal distribution, $\varepsilon_v \sim \mathcal{N}(0, \sigma_v)$, where the standard deviation $\sigma_v$ is calculated by

$$\sigma_v^2 = \frac{p - p^2}{v}, \quad p = (1 - \theta_2 v^{\theta_1}) + c(\theta_2 v^{\theta_1}). \tag{8}$$

The second term $\varepsilon_M(v)$ is the variance in overall accuracy. It represents the classifier's differences in performance, even if trained and evaluated on the same data, due to random decisions in the training process. We model this error-term to follow a normal distribution $\varepsilon_M(v) \sim \mathcal{N}(0, \sigma_M(v))$, where the standard deviation follows a power law:

$$\sigma_M^2(v) = b \cdot \alpha(v)^d. \tag{9}$$

Here $\alpha(v) = \theta_2 \cdot v^{\theta_1}$ is the power law portion of Equation 7. The values for $b$ and $d$ are determined by examining the actual variance in model losses when trained on LSUN or ImageNet (see Section 5). From observing these we saw that a power law between accuracy and variance achieves a good fit and determined $b$ and $d$ from it. The resulting values are

$$d = -10\theta_1 \cdot (\theta_2 v_*^{\theta_1}), \quad b = 0.0018 \cdot (\theta_2 v_*^{\theta_1}). \tag{10}$$

Here volume $v_* = 100,000$ represents the target volume of the simulated learning curves. Finally the volumes $v_0$ and $v_s$ are defined by ensuring continuity of the resulting learning curve when ignoring error terms. They are

$$v_0 = \left(\frac{1 - c}{\theta_2}\right)^{-\frac{1}{\theta_1}}, \quad v_s = \left(\frac{\delta}{\theta_2}\right)^{-\frac{1}{\theta_1}}. \tag{11}$$

With a model for learning curves, we can simulate the performance of $\mathbb{S}$-trained models including the variance terms from above. For these we emulate a dataset of 20 different classes and a total training volume of 100K samples. We created 12 different learning curves and evaluated them (with variance terms) 5 times on each $x_i$ to perform the sampling strategies as described in the previous section. As learning curve shape parameters we chose discrete values for $\theta_1$, $\theta_2$, and $\delta$ to produce learning curves with many crossings (see the right panel of Figure 1 for a plot of the learning curves without their error terms).

## 5 DATASETS AND MODELS EVALUATED

Outside of simulated learning curves, we have also trained several models on datasets to validate our strategies in real scenarios. For this we have selected three datasets with different characteristics. ImageNet (Deng et al. (2009)) represents a well studied large dataset that has many different classes.

---

[3] We change the notation of training volumes from $x$ to $v$ in this section to remove ambiguity to the notations introduced in the previous section

Table 2: Datasets used

| Dataset | Target Volume | Number of Classes selected |
|---|---|---|
| ImageNet Deng et al. (2009) | 165,712 images | 7 |
| LSUN Yu et al. (2015) | 146,394 images | 6 |
| Plant Dataset Beck et al. (2022) | 45,000 images | 9 |

**Time and loss costs for choosing $k$ models sampling 1 volume**

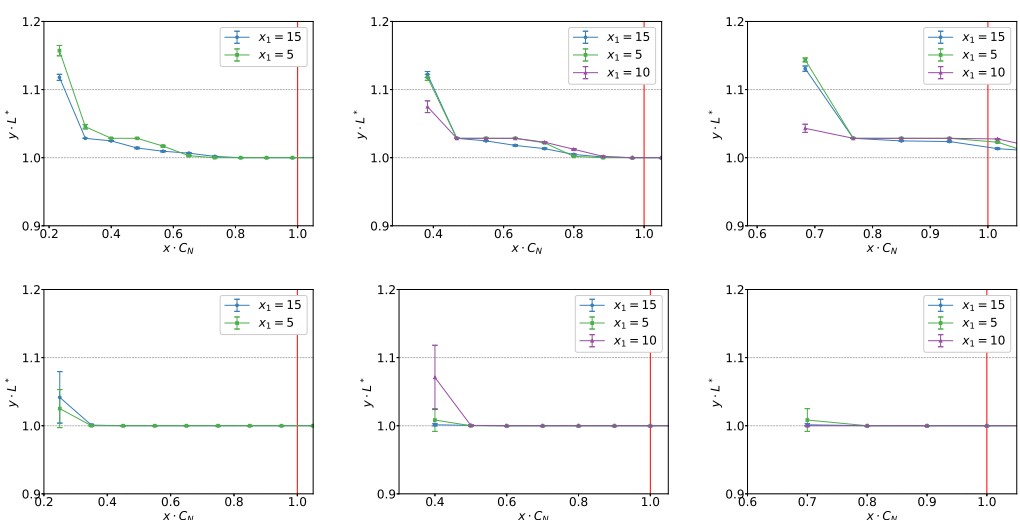

Figure 2: Time and loss costs for LSUN (first row) and ImageNet (second row) when sampling from a single data volume $x_1$ only. The first column represents a value of $C_s = 0.15C_N$. The second and third column represent $C_s = 0.3C_N$ and $C_s = 0.6C_N$, respectively.

LSUN (Yu et al. (2015)) has fewer classes and thus a higher image density per class. Finally, we also retrieved a plant dataset through a data portal (Beck et al. (2022)). The such constructed dataset is much smaller than ImageNet or LSUN, but offers different characteristics. Its images show different plants on front of a blue background and each individual plant is imaged from different angles. Thus, the images contain a lot of similar features and can be extremely similar to each other. Table 2 gives an overview on the different dataset characteristics and the models applied to them.

## 6 RESULTS

To simulate the usage of learning curves from a practitioner's perspective we used the following approach. First, according to the training parameters laid out above we repeatedly trained each model class on each training volume and tracked the resulting training time and OOD accuracy for each training run. This resulted in a pool of actual model performances from which we can sample. Then, after determining the number of repetitions $l_{i,j}$ for a given strategy we sample accordingly from this pool. Consequently, the resulting learning curve fits are dependent on which samples had been picked. This is in accordance with the randomness a practitioner is faced with when they want to use learning curve estimations. Since each execution of an individual strategy can yield different results, we perform each strategy 30 times, and report in the following mean results and standard deviations.

### PERFORMANCE COSTS

Once learning curves are fitted only $k$ models are selected according to the predictions $\hat{\eta}_j(x_N)$ (say, $M_{j_1}, \ldots, M_{j_k}$). Define the loss of that model by $L_{\text{found}} = \min_{j=j_1,\ldots,j_k} \{y_{N,j}\}$. If we define by

**Time and loss costs for choosing $k$ models sampling 2 volumes**

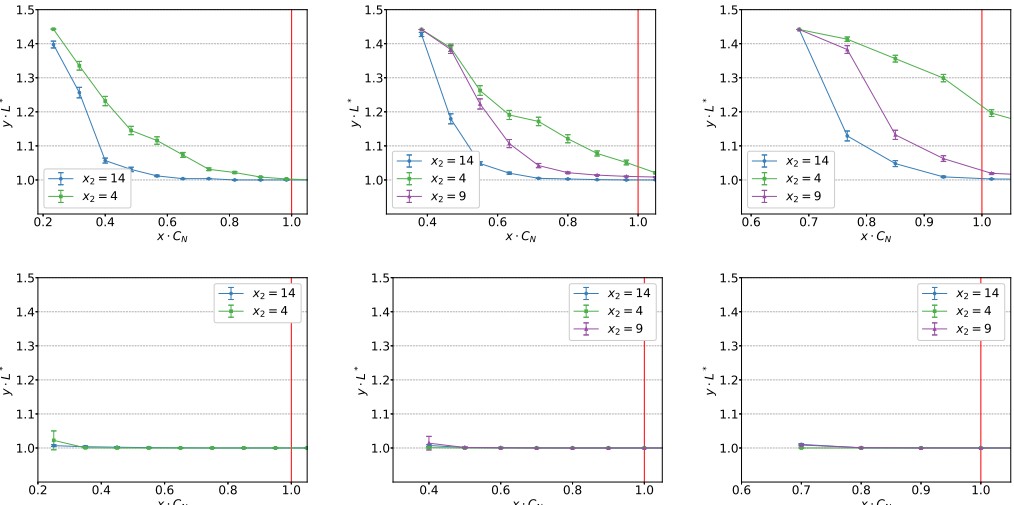

Figure 3: Time and loss costs for LSUN (first row) and ImageNet (second row) when sampling from two data volumes with $x_1$ fixed at $0.01x_N$. The first column represents a value of $C_s = 0.15C_N$. The second and third column represent $C_s = 0.3C_N$ and $C_s = 0.6C_N$, respectively.

$L^* = \min_j\{y_{N,j}\}$ the best loss of the $\mathbb{S}_N$-trained models, we have $L_{\text{found}} = y \cdot L^*$ for some $y \geq 1$. The difference between the two quantities is the hit in performance we have to suffer for using only $C_s + C_t$ training costs. If our prediction strategy is sound, $L_{\text{found}}$ will be very close or equal to $L^*$.

CHOICE OF VOLUMES

We investigate now how the choice of volumes influences $L_{\text{found}} = y \cdot L^*$. We represent these results by plotting the multiplier $y$ against the cost savings $x$ defined in Equation (5). In these plots the costs $C_N$ are represented by the red line at $x = 1$ and the best obtainable loss by the line through $y = 1$. Thus, we can easily compare each strategy to the default of just training every model on the target volume, by seeing where the plotted point lies in relation to the coordinates $x = 1, y = 1$. Each series on these plots represents one sampling strategy (determining $C_s$) and each point represents one choice of $k$ for that strategy (determining $C_t$).

We first consider sampling from a single volume of size $x_1 = 0.15x_N$, $x_1 = 0.1x_N$, and $x_1 = 0.05x_N$, respectively. By varying the amount of samples drawn we can enforce $C_s = 0.6C_N$, $C_s = 0.3C_N$, or $C_s = 0.15C_N$. For example, to achieve a $C_s = 0.6$ we sample 4 times on 15% of the training volume or sample 12 times on 5% of the training volume. The results for training on the different training volumes and the different training sets (ImageNet and LSUN) are presented in the panels of Figure 2. In general, we see that the choice of volume (different lines in each plot) does not have a large impact on $y \cdot L^*$ in any scenario. We see, however, that a small allocation of $C_s$ is generally beneficial, as more models can be evaluated on the target volume without exceeding $C_N$ and the best model can be found (curves reach the $y = 1$ line). In the appendix sampling single volumes for the simulated learning curves is discussed.

Next we sample from two volumes, where one volume is fixed at $x_1 = 0.01x_N$ and the other is either $x_2 = 0.14x_N$, $x_2 = 0.09x_N$, or $x_2 = 0.04x_N$. Again using $C_s = 0.6C_N$, $C_s = 0.3C_N$, or $C_s = 0.15C_N$ the models will be sampled the same amount of times as when we sampled one training volume. The subfigures of Figure 3 present the results. While for ImageNet the difference in sampling one or two data volumes is marginal, the performance on LSUN is very different. We can see that sampling two data volumes leads to initially higher losses of roughly $1.4 \cdot L^*$ compared to the losses of less than $1.2 \cdot L^*$ when sampling only a single data volume. This means the $k$ chosen models perform worse for small values of $k$. However, the loss costs also decrease faster and at $k = 5$ are lower compared to sampling only one training volume, if $x_2 = 0.14x_N$. Additionally, a

**Time and loss costs for choosing $k$ models sampling 4 volumes**

Figure 4: Time and loss costs for LSUN (first row) and ImageNet (second row) when sampling from four data volumes uniformly. The first column represents a value of $C_s = 0.20C_N$. The second and third column represent $C_s = 0.3C_N$ and $C_s = 0.6C_N$, respectively.

wider spread between $x_1$ and $x_2$ gives a better prediction performance overall. This trend can also be observed for the Plant Dataset and when predicting the simulated learning curves (see Appendix).

Finally, we sample from four different volumes to estimate learning curves using a non-linear least square method. For this we define three sequences of volumes of four volumes each. The first is $x_1 = 0.01x_N$, $x_2 = 0.04x_N$, $x_3 = 0.08x_N$, and $x_4 = 0.16x_N$, whereas the second sequence is $x_i = 2^i \cdot 0.01x_N$ and the third sequence is $x_i = i \cdot 0.01x_N$. Thus, the first sequence emphasises larger training volumes and the third sequence emphasizes smaller training volumes. We see that for the LSUN dataset the derived learning curves are not sufficient to find the best performing model before reaching a cost of $C_N$ and that the sequence that emphasizes larger training volumes leads to better predictions. For the ImageNet dataset we only observe that the best model is found very early in all configurations, similar to the results we get from sampling on two volumes.

## 7 CONCLUSION

In this paper we have formulated the problem of how to sample models on smaller training volumes for the purpose of predicting performance, when trained on large training volumes. We have evaluated several scenarios in which deep convolutional neural networks are used to label image data for different sampling strategies. Overall, we made the following observations: (1) Sampling from more than one volume to obtain a learning curve fit leads to better performance prediction compared to sampling only a single volume (which does not allow for construction of a meaningful learning curve). (2) The benefits of sampling from more than two volumes are negligible, at least in the scenarios we have investigated. (3) When deciding which two (or more) volumes to sample for fitting learning curves a wide spread of volumes leads to better performance prediction. (4) Sampling volumes more often (to get a better estimate on the mean performance of the model when trained on that volume) is generally less beneficial than using that training time to increase the number of selected models $k$.

Further investigation into sampling strategies should be performed. Logical next steps would be (1) considering a wider scope of application scenarios; (2) considering sampling from additional numbers of volumes; (3) considering sampling strategies that are sample specific volumes more often than others, i.e. $l_{i,j}$ can be different for differing $i$ or $j$.

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

Table 3: Models used

| **Models** |
| --- |
| DenseNet121, DenseNet169, DenseNet201 Huang et al. (2016) |
| EfficientNetB7 Tan & Le (2019) |
| EfficientNetV2S Tan & Le (2021) |
| InceptionV3 Szegedy et al. (2015) |
| InceptionResNetV2 Szegedy et al. (2016) |
| InceptionV3 Szegedy et al. (2015) |
| MobileNet Howard et al. (2017) |
| MobileNetV3Large, MobileNetV3Small Howard et al. (2019) |
| NASNetLarge, NASNetMobile Zoph et al. (2017) |
| ResNet50, ResNet101, ResNet152 He et al. (2015) |
| ResNet50V2 He et al. (2016) |
| ResNetRS50 Bello et al. (2021) |
| VGG16, VGG19 Simonyan & Zisserman (2014) |
| Xception Chollet (2016) |

## A  APPENDIX

Table 3 presents a list of models used in the evaluation described in Section 6.

**Time and loss costs for choosing $k$ models sampling 2 volumes**

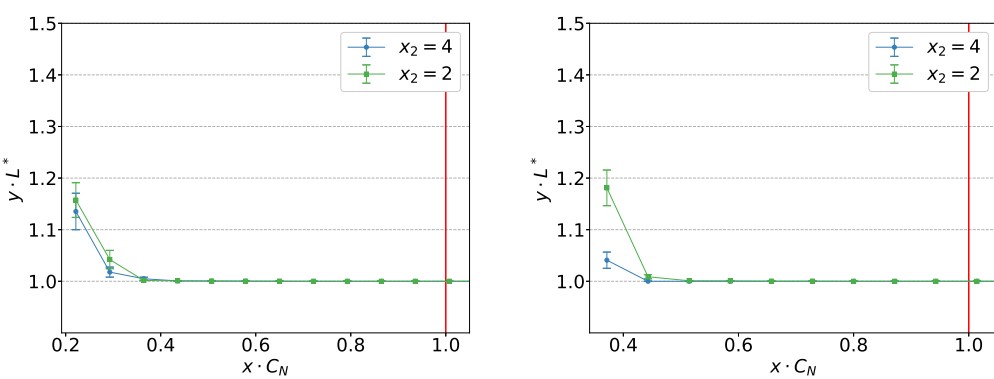

Figure 5: Time and loss costs for the plant dataset when sampling from two data volumes with $x_1$ fixed at 1% of the full training dataset, with $C_s = 0.15C_N$. (left) and $C_s = 0.3C_N$ (right) respectively.

## EVALUATION ON SIMULATED LEARNING CURVES

### SAMPLING ONE VOLUME

We can see in the first row of Figure 6 the loss and time costs for sampling a single volume on the simulated learning curves. Here . Since, the learning curves had been created purposefully to exhibit many crossings of curves, we see much higher values for $y$ in general. We can also see that sampling from larger volumes leads to better results (when $x_1 = 0.15x_N$). However, sampling too large volumes also seems to be detrimental, as can be seen for $x_1 = 0.45x_N$ in the third panel.

### SAMPLING TWO VOLUMES

Also in Figure 6 we can see how the situation changes for predicting the simulated learning curves, when two volumes are sampled. Overall lower values for $y$ can be achieved before the costs exceed

**Time and loss costs for choosing $k$ models sampling one and two volumes**

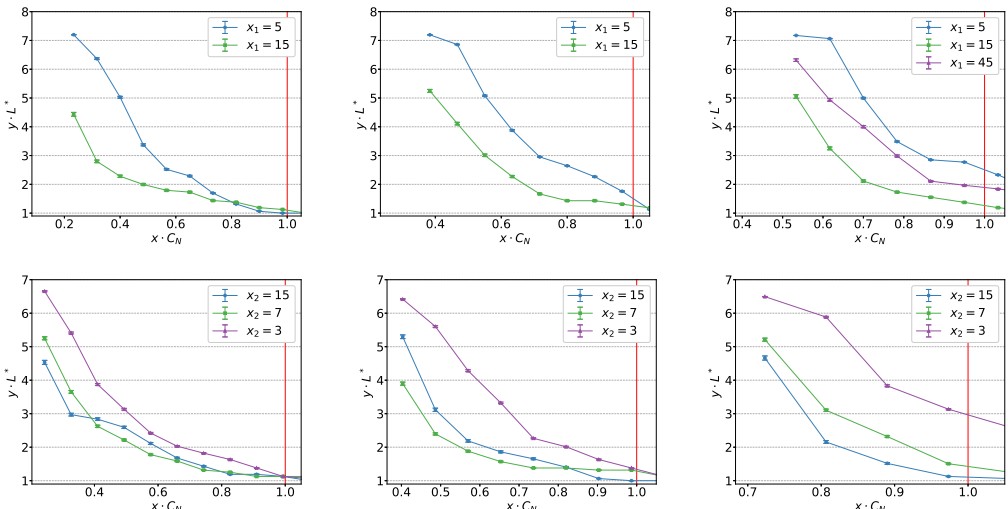

Figure 6: **First row:** Time and loss costs for predicting the performance of the simulated learning curves when sampling from a single data volume $x_1$ only. The first column represents a value of $C_s = 0.15C_N$. The second and third column represent $C_s = 0.3C_N$ and $C_s = 0.45C_N$, respectively. **Second row:** Instead sampling from two volumes with the first volume fixed at 1% of the target volume. The first column represents $C_s = 0.16C_N$. The second and third column represent $C_s = 0.32C_N$ and $C_s = 0.64C_N$, respectively

**Time and loss costs for choosing $k$ models sampling four volumes**

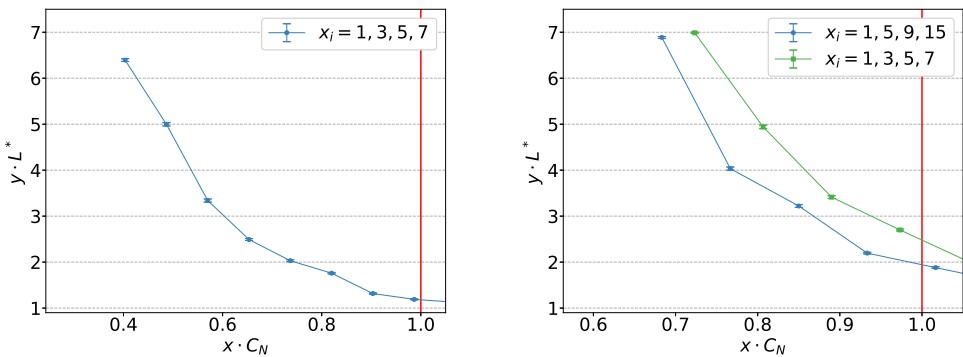

Figure 7: Time and loss costs for sampling four volumes, when predicting the simulated learning curves.

$C_N$. We also see that sampling from a wider spread is generally beneficial for predicting which models will perform well when trained on the target volume.

SAMPLING FOUR VOLUMES

In Figure 7 we can see four volumes sampled. The values for $C_s$ are $0.6C_N$ and $0.64C_N$, respectively for the two series in the second panel, and $0.32C_N$ for the first panel. The volumes sampled in both panels are $x_1 = 0.01x_N$, $x_2 = 0.03x_N$, $x_3 = 0.05x_N$, and $x_4 = 0.07x_N$, the second panel also shows a series for volumes $x_1 = 0.01x_N$, $x_2 = 0.05x_N$, $x_3 = 0.09x_N$, and $x_4 = 0.15x_N$ (which corresponds to $0.6C_N$). We can see that sampling from four volumes does not lead to finding the best models faster, compared to sampling from two volumes. However, we still see a trend that spreading the volumes over a wider range leads to better estimates, as we have seen for sampling

from two volumes and as we have also seen for the LSUN dataset. This matches our intuition that sampling from a wider range should lead to a better estimate of the learning curves slope parameter, which is crucial for identifying which learning curves will cross in for larger training volumes.

