# OpenReview forum: "Strategies and impact of learning curve estimation for CNN-based image classification"
_ICLR.cc/2024/Conference — Submitted to ICLR 2024_

### Official Review · Reviewer_KbgV · 2023-11-01

**Soundness:** 2 fair
**Presentation:** 2 fair
**Contribution:** 2 fair
**Rating:** 3
**Confidence:** 2

**Summary:**

The paper evaluates several approaches to fit learning curves of models trained on various subsets of data to predict the model's final performance. Based on the evaluation the paper makes several suggestions about the best way to maximize the accuracy of the predicted learning curves given a specific compute budget.

**Strengths:**

Predicting learning curves is an interesting topic and has the potential to save costs for hyperparameter optimization and/or model training in general. Having some comprehensive exploration about the best approaches to predict learning curves is, therefore, interesting and important.

**Weaknesses:**

Overall I found the paper a bit hard to understand as there is a lot of notation that is hard to follow (e.g., C_s, C_t, C_N, etc). I think it's also confusing to call the loss on the test set an out-of-distribution (OOD) loss, as the test set (usually) comes from the same domain/distribution as the training set and is not OOD.

The current evaluation is also limited to only LSUN and ImageNet (and one more plant dataset though I can't find the evaluation for that) and only for discriminative tasks. It would be helpful to scale this to other tasks, datasets, and model architectures to see if the findings stil hold for those cases.

**Questions:**

I am not an expert in this topic so maybe I missunderstood things but I honestly don't really understand the graphs in Figures 2, 3, etc. I understand the y-axis is the difference in predicted loss vs actual loss for a given model (so should be as close to 1 as possible) but I don't understand the x-axis, how the value of C_s relates to the graphs, and what the individual sampling points in each graph correspond to.

As such, it's also difficult for me to understand the main message of the paper, e.g., I also don't understand where you compare the different model architectures you're sampling from and how the learning curves correspond to those models.

---

### Official Review · Reviewer_HBcf · 2023-11-03

**Soundness:** 1 poor
**Presentation:** 2 fair
**Contribution:** 1 poor
**Rating:** 3
**Confidence:** 2

**Summary:**

The paper investigates strategies for estimating learning curves in order to predict model performance when trained on large datasets. The main contributions are:

- Proposes a framework and notation for fitting learning curves from performance samples on small training volumes.

- Introduces a model to simulate learning curve behavior and generate synthetic results. The model captures the three key regions of learning curves: the small data region, power law region, and irreducible error region.

- Evaluates several sampling strategies on the simulated learning curves as well as three real image classification datasets. Strategies differ in number of volumes sampled, volume sizes, and sampling frequencies.

- Finds that sampling two volumes with a wide spread provides a good tradeoff between accuracy and cost. Sampling more volumes or sampling volumes more often does not improve results much. A wider spread of volumes leads to better performance prediction.

- Overall, shows that sampling models on small subsets can produce learning curve fits that allow predicting the best performing model on the full dataset while reducing overall training time. The paper examines how to do this sampling efficiently.

In summary, the paper introduces the concept of learning curve estimation for model performance prediction and systematically studies different sampling strategies for this task. The simulated learning curve model and analysis on real datasets provide insights into efficient learning curve sampling.

**Strengths:**

Fitting learning curves from samples is not new, but the explicit focus on sampling strategies might represent an original angle, of which I am not sure.

**Weaknesses:**

Here are some potential weaknesses and suggestions for improvement:

- I do not see much fundamental difference between the data collection strategy in Mahmood et al. (2022b) and subset sampling in the paper. Isn't collecting more data similar to using a subset of a bigger set and using more later in rounds?

- The conclusions about optimal sampling strategies could be better quantified, rather than relying on interpreting the graphs. Statistical tests or performance indexes could help concretize the differences.

- Only image classification with CNNs is considered. Expanding the evaluation to other domains (NLP, speech, etc) and model types would strengthen generalizability.

- Additional sampling strategies could be proposed and evaluated, such as non-uniform sampling frequencies or active learning approaches. This may uncover better approaches.

- The simulated learning curve model relies on several parameters. Sensitivity analysis on these parameters could be insightful.

- The costs are defined abstractly in terms of training volume. Incorporating actual runtimes/FLOPs could give more practical insights.

- The power law learning curve form is assumed without much justification. Discussion of limitations or comparisons to other forms could add depth.

Overall, while the paper introduces a useful framework, the limited gains shown so far temper the impact. Expanding the empirical analysis and strengthening the conclusions would help in assessing the viability of the approach. Please let me know if you would like me to elaborate on any of these suggestions or if you have any other specific guidance on improving the constructive critical analysis. I appreciate you taking the time to help me provide meaningful feedback.

**Questions:**

As mentioned above in the weakness part, I do not see a novelty in the proposed strategy compared to data collection in rounds.
A justification of novelty and originality of the proposed strategy would be appreciated.

---

### Official Review · Reviewer_qb3L · 2023-11-07

**Soundness:** 2 fair
**Presentation:** 1 poor
**Contribution:** 2 fair
**Rating:** 3
**Confidence:** 2

**Summary:**

This study explores the neural scaling principles for model selection for a visual tasks. The approach introduced in this research involves the selection of hyperparameters through training a neural networks on smaller portions of the complete training dataset, and leveraging the neural scaling law to determine the most effective hyperparameters for larger-scale training, while managing the overall training costs.

The effectiveness of this selection method is assessed across vision datasets, including ImageNet and LSUN.

**Strengths:**

- The paper investigates the important problem of hyperparameter selection in a cost effective manner.
- Empirical results demonstrate that, on LSUN and ImageNet, the author can attain nearly optimal model performance with just half the training budget for hyperparameter selection.

**Weaknesses:**

- One of the primary contributions of the paper is the utilization of the neural scaling law for hyperparameter selections. However, it appears that when using a single subset (i_1) on ImageNet, most of the model selection performance is achieved. This finding does not support the importance of employing the neural scaling law in this case since the selection mechanism relies solely on validation performance and does not estimate the neural scaling laws.

- I believe that the clarity of the paper could be enhanced to better describe its main contributions and the technical implementation and evaluation of these contributions. Specifically:
    - In general, it seems that the primary objective of neural scaling is to predict the performance as data scales and use this information for model selection at a smaller scale. Therefore, I am not entirely clear about what the paper's main contribution is. Is it a specific instantiation of model selection using neural scaling law designed for visual tasks?
    - It's unclear why the paper reports out-of-distribution (OOD) accuracy when it appears to come from the same distribution as the training data.
     - In the experimental section, more information is needed regarding the actual performance achieved on ImageNet and other datasets, as well as details about the range of hyperparameters explored. Additionally, there is mention of results for the Plant dataset in Section 5, but the paper only presents results for ImageNet and LSUN. Is the synthetic dataset used in the experimental section?

These suggestions could help improve the paper's clarity and better convey its main contributions and evaluation process.

**Questions:**

See weaknesses. Overall, I would encourage the author to clarify what are the paper main contributions and their novelty.

---

### Official Review · Reviewer_rA2E · 2023-11-08

**Soundness:** 3 good
**Presentation:** 3 good
**Contribution:** 3 good
**Rating:** 6
**Confidence:** 3

**Summary:**

The paper introduces the concept of fitting learning curves from samples on small training volumes and proposes various strategies for doing so efficiently. Learning curves describe how a machine learning model's performance improves with more training data, and they can be used to predict which models will perform best on a full dataset, ultimately saving training time. The paper evaluates these strategies using simulated learning curves and experiments with popular datasets and image classification models.

The work builds upon previous research on neural scaling laws and learning curves in deep learning, with a focus on practical strategies for estimating learning curves. It also addresses the challenge of choosing subset sizes and how often to sample models to obtain accurate learning curve estimates while reducing training time.

The results show that sampling from more than one volume for learning curve fitting leads to better performance prediction, and a wide spread of volumes is beneficial for accurate predictions. However, sampling volumes more often does not provide significant benefits compared to increasing the number of selected models. The paper suggests further investigation into sampling strategies and their application in different scenarios.

Overall, the paper presents a valuable contribution to the field of deep learning and learning curve estimation, offering insights into efficient strategies for model selection and training time reduction.

**Strengths:**

This paper presents a novel and valuable contribution to learning curve estimation. There are several notable strengths and points of novelty in this work:
* Introduction of Learning Curve Fitting from Small Training Volumes: The paper introduces the concept of fitting learning curves from samples on small training volumes. This is a novel approach that can significantly reduce the time and resources required for model selection and training.
* Practical Application: The paper focuses on practical strategies for estimating learning curves, which is highly relevant in real-world machine learning scenarios. This practicality distinguishes it from some earlier theoretical works on learning curves.
* Efficient Sampling Strategies: The work proposes and evaluates various sampling strategies for learning curve estimation. It provides insights into how to choose subset sizes and how often to sample models efficiently. This can be of great help to practitioners looking to save time and resources in model selection.
* Comprehensive Evaluation: The paper conducts a thorough evaluation of the proposed strategies. It includes experiments with popular datasets and image classification models, as well as simulations to test the strategies in a wider scope. This comprehensive evaluation strengthens the practical applicability of the findings.
* Comparison with Previous Research: The paper places itself in the context of previous works on learning curves and neural scaling laws. It highlights the differences and complementary aspects of this work in comparison to earlier research. This contextualization helps reviewers and readers understand where this work fits within the existing literature.
* Clear and Well-Structured Presentation: The paper is well-structured and clearly presents the problem, methodology, and results. This makes it accessible to a wide audience, including both researchers and practitioners.

In summary, the strengths and novelty of this work lie in its practical approach to learning curve estimation from small training volumes, its efficient sampling strategies, and its thorough evaluation. These contributions make it a valuable addition to the field, addressing the needs of machine learning practitioners seeking to optimize model selection and training.

**Weaknesses:**

While the paper presents a novel approach to learning curve estimation from small training volumes and offers valuable insights, there are some weaknesses and limitations that should be addressed:
* Assumptions in Learning Curve Modeling: The paper assumes a power-law relationship for learning curves, which may not always hold true for all types of machine learning models. It would be valuable to discuss the limitations of this assumption and explore how the proposed methodology performs when the power-law relationship does not hold.
* Sensitivity to Dataset and Model Choice: The evaluation is conducted with a specific set of convolutional neural network architectures and datasets. It's important to acknowledge that the effectiveness of the proposed strategies may vary with different datasets and model types. A more extensive evaluation on a wider range of datasets and model architectures would strengthen the generalizability of the findings.
* Statistical Significance of Results: The paper mentions conducting each strategy 30 times and reporting mean results and standard deviations. However, it would be useful to perform statistical tests to determine the significance of the differences observed between strategies and to provide confidence intervals for the reported results.

**Questions:**

The paper is well written with good practical contributions to learning curve estimation. I have mentioned some concerns in the weaknesses section which I would like the authors to talk about or look into.

---

### Meta-Review · Area_Chair_dRSe · 2023-12-05

**Metareview:**

The submission proposes to use performance on subsets of training data with multiple models, and neural scaling laws to extrapolate learning curves for model selection.  While the approach addresses and important problem, the results and presentation were not convincing to a majority of reviewers, and the authors did not provide a rebuttal.

**Justification For Why Not Higher Score:**

Concerns from a majority of reviewers and no rebuttal provided.

**Justification For Why Not Lower Score:**

N/A

---

### Decision · Program_Chairs · 2024-01-16

Reject